# OpenReview forum: "ImmersePro: End-to-End Stereo Video Synthesis Via Implicit Disparity Learning"
_ICLR.cc/2025/Conference — ICLR 2025 Conference Withdrawn Submission_

### Official Review · Reviewer_uxuU · 2024-10-17

**Soundness:** 2
**Presentation:** 1
**Contribution:** 2
**Rating:** 3
**Confidence:** 5

**Summary:**

This paper propose an implicit disparity map to avoid the texture artifact problem occurred in explicit disparity and a layered disparity selection module to improve the result. Then the authors collect a new stereo video dataset to assist stereo video generation. Some self-attention, cross-attention and context fusion modules are added to enrich the paper. The extensive experiments show they achieve great spatial and temporal consistency in stereo video.

**Strengths:**

1. The proposed implicit disparity and layered disparity improve the results.

2. This paper collect a new stereo video dataset.

**Weaknesses:**

1. The problem of existing methods is unclear. As shown in Line 051-052, what drawbacks? The authors should give detailed discussion and show the visualization or experiment at the beginning to better understanding.

2. Vertical distortion in the dataset. In section 3 Line 180-181, the authors say that the performance of RAFT-Stereo is worse than RAFT, which does not align with intuition as RAFT-Stereo is specifically designed to implement horizontal offsets for stereo matching. The probable reason is that the proposed dataset contains vertical offset, which is not suitable for stereo video synthesis. The author should do more analysis about the proposed dataset to show the accuracy of their dataset.

3. The presentation of the method section is poor.

       a) Line 348 Eq. 3, typo in equation.
       b) Line 351, what is $D_{i,j}^d$
       c) How the dimension D (7) is obtained.
       d) Why the operation of layered disparity can be used to select the layer. The authors should give the illustration in detail (not just an algorithm).
4. Question about the novelty. The disparity index selection is not a novel operation as it is widely used in stereo matching Fast-ACV [1], the authors should clarify the novelty of their layered disparity compared with the one used in stereo matching. What's more, why implicit disparity works better than explicit disparity, as in operation, the difference is just the turn (calculating the disparity by weighted-summation first and using warp function (explicit), using warping function first and calculating the image by weighted-summation). The authors should give more theoretical analysis and high-level insight to validate the motivation.

[1] Xu, G., Wang, Y., Cheng, J., Tang, J., Yang, X. (2023). Accurate and efficient stereo matching via attention concatenation volume. IEEE Transactions on Pattern Analysis and Machine Intelligence.

5. Lack experiments.

       a) Lacking comparison with explicit disparity. The explicit disparity can also be used and the disparity index selection can be applied, this experiment is vital for showing the motivation of this paper.
       b) Lacking comparison with recent SOTA methods. This paper in Line 051-053, 135-136 claims that warping-inpainting based methods work bad. However, they does not compare with this methods which weakens the contribution of this paper. The author should add comparison with following SOTA methods [2], [3], [4]. Note that although these methods do the task of stereo image generation from mono, the performance are great when directly applying to video.
[2] Tucker, R.; and Snavely, N. 2020. Single-view view synthesis with multiplane images. In Proceedings of the IEEE/CVF Conference on Computer Vision and Pattern Recognition, 551–560.

[3] Han, Y.; Wang, R.; and Yang, J. 2022. Single-view view synthesis in the wild with learned adaptive multiplane images. In ACM SIGGRAPH 2022 Conference Proceedings, 1–8.

[4] Wang, X.; Wu, C.; Yin, S.; Ni, M.; Wang, J.; Li, L.; Yang, Z.; Yang, F.; Wang, L.; Liu, Z.; et al. 2023b. Learning 3D Photography Videos via Self-supervised Diffusion on Single Images. arXiv preprint arXiv:2302.10781.

**Questions:**

See the weaknesses.

---

### Official Review · Reviewer_N7bR · 2024-11-02

**Soundness:** 2
**Presentation:** 2
**Contribution:** 2
**Rating:** 3
**Confidence:** 4

**Summary:**

This paper presents both a new dataset for mono to stereo video conversion based on permissively licensed stereo youtube videos as well as a method for performing the conversion automatically.  The authors claim the following contributions:
* YouTube-SBS dataset which includes 7 million stereo pairs from over 400 videos.
* ImmersePro - a mono to stereo conversion method

The authors also present a number of qualitative and quantitative experiments validating the approach.

**Strengths:**

* Timely problem
* Method is clear and intuitive
* YouTube SBS dataset preprocessing is a significant contribution
* Results look good

**Weaknesses:**

* Figure 3 rows 2 and 3 illustrating implicit disparity and the associated warped image looks more like a bug rather than a valid result.  Is it the case that row 3 was obtained by training a network with layered disparity included, but then omitted in the final rendering?  Or does it represent a model where layered disparity was excluded from both training and rendering?  Row 3 shows more than just "blurring" artifacts as suggested in section 4.4 and could impact the ablations in table 3.

* Two of the baselines are pretty old at this point and the proposed method is making use of Depth Anything, which presumably offers quite an advantage.  I think a very simple baseline would be to use Depth Anything v2 on the left view and then forward warp it to produce the right view.  How the holes from the forward warp are filled in is an open question, but you could generate a mesh for warping to make sure all pixels are filled.

* There are no metrics related to temporal consistency.  I'd recommend reading Luo, et.al, "Consistent Video Depth Estimation" SIGGRAPH 2020 to get an idea on how to illustrate temporal consistency.

* Section 6 paragraph 1 raises concerns about the metrics used to compare the proposed method and baselines.  Many of these methods do indeed need to be treated as "scale invariant" in one way or another (whether it be the focal length, IPD, or something else).  Was there care taken when doing comparisons with the baselines to ensure a scale was selected that best minimized the error reported in the metrics?

* The "GT" presented in figure 5 clearly should not be interpreted as GT.  The predicted L2R disparity also appears to be well below the bar for modern disparity prediction.

* I think if the authors want to show GT disparity, then they should use a dataset like MPI Sintel.  They could also use this for the temporal consistency evaluation.  Reading Kopf, et.al. "Robust Consistent Video Depth Estimation" CVPR 2021 might be useful here.

**Questions:**

* Was the YouTube-SBS dataset filtered to include only videos with public domain and CC-BY license or does it also include content with the standard youtube license?  Can you give a breakdown of percentage of each license?

**Details Of Ethics Concerns:**

A significant portion of the contribution is a YouTube dataset.  The authors state "To ensure accessibility for future research, we select videos that (1) have existed for at least one year, and (2) from accounts that have at least 500 followers."  However, they do not state whether they filtered videos by license and whether or not the dataset can be stored and redistributed in perpetuity.

---

### Official Review · Reviewer_ozgs · 2024-11-03

**Soundness:** 2
**Presentation:** 2
**Contribution:** 2
**Rating:** 3
**Confidence:** 3

**Summary:**

The paper presents a framework for converting single-view videos into stereo videos using a dual-branch architecture ((1) disparity branch to extract disparity-oriented feature maps, (2)  context feature extractor to extract contextual semantic features. ) with spatial-temporal attention. This approach leverages implicit disparity guidance to generate stereo pairs without the need for explicit disparity maps, thereby reducing errors. Additionally, the authors introduced the YouTube-SBS dataset, which includes 423 stereo videos and over 7 million stereo pairs, for training and benchmarking purposes. Experimental results demonstrate that ImmersePro enhances the quality of stereo videos compared to existing methods.

**Strengths:**

+ The introduction of a dual-branch architecture and implicit disparity guidance for stereo video generation
+ Introduction of the YouTube-SBS dataset
+ Ablation studies on various modules of the framework

**Weaknesses:**

- Lack of complexity analysis and comparison with other methods, such as “stereo-from-mono.”
- Ablation studies do not provide conclusive evidence of the necessity of specific modules. For instance, the ablation on context fusion shows a PSNR drop of less than 0.2 dB.
- The paper lacks sufficient novelty in theoretical contributions or architectural design.
- The main figure on architecture is not very readable; a zoom-in on implicit disparity (as the contribution) would help in understanding the architecture.
- The complexity of the stack of multiple horizontally shifted copies is not discussed, and it is not explained why only horizontal shifts are sufficient for prediction.
- Evaluation on other benchmark datasets is not shown.

**Questions:**

1) What is the computational complexity of predicting a view in the proposed framework? Is there any latency?
2) Why does the method fail to converge when “implicit disparity” is removed? Can this component be replaced with an alternative approach?

---

### Official Review · Reviewer_7srE · 2024-11-04

**Soundness:** 3
**Presentation:** 3
**Contribution:** 3
**Rating:** 6
**Confidence:** 5

**Summary:**

This paper introduces a method to convert single-view videos into stereo videos. The authors propose a dual-branch architecture with spatial-temporal attention layers, eliminating the need for explicit disparity maps. The authors also build a large scale stereo video dataset, containing 7M stereo frames from 423 videos. The experiments demonstrates the effectiveness of the method and dataset.

**Strengths:**

1. These paper introduces a large dataset for stereo video generation, which will benefit the research in this area.

2. The design for the implicit disparity is reasonable, which is also consistent with the findings from Vimeo that the direct optical flow is not good for image systhesization.

**Weaknesses:**

1. The dataset details are missing.

(a) How many frames per video? Does the videos have the same length? If not, what's the length distribution?

(b) Stereo videos often contain multiple scenes. These video clips are not consistent. The author only measured the spatial consistency between the left and right view. However, it seems that the author missed the temporal consistency, which is important for the stereo videos. How does the author handle it?

(c) In Eq. 1, the authors leveraged the optical flow estimated by RAFT to measure the error. However, optical flow is not accurate in the stereo pairs as it containing the movement in the vertical direction. The author should only consider the horizential direction. Although authors claimed that "We also evaluated STTR (Li et al., 2021) and RAFT-Stereo (Lipson et al., 2021), but these two methods produced worse results.", the aurhors should find a more accurate disparity estimation method or constrain RAFT in the horizential direction.

2. Experiments

(a) For a fair comparison, the authors are required to re-train or finetune the state-of-the-art methods on the new datasets.

(b) To check the effectiveness of the proposed dataset, the authors need to evaluate image-based models trained on the new dataset on the previous image-based dataset.


Minor

1. "6.39%(SSIM), and 5.10% (PSNR)" is not professional. Most of the time, we only report the differences, e.g. 1.1 dB in term of PSNR. As PSNR is based on the log function, percentage is not correct.

**Questions:**

Please refer to the weakness section.

---

### Note · Authors · 2024-11-15

I have read and agree with the venue's withdrawal policy on behalf of myself and my co-authors.